# From Denoising to De-Channeling: Integrating Physical Channel Priors into Diffusion Models for Radio Signal Understanding

Yaoqi Liu [1 2]   Jin Wang [1]   Chunchen Wang [2]   Hui Wang [1]   Chuan Shi [1 2]

## Abstract

In recent years, wireless signal recognition (WSR), which leverages artificial intelligence (AI) to identify properties of passively received radio signals, has garnered significant attention due to its broad applications, such as spectrum management. Existing WSR methods typically learn directly from received signals, which are distorted by physical wireless channel effects such as fading, and current denoising diffusion models lack de-channeling capabilities, which leads to performance degradation. Therefore, we propose PWC-Diff, a novel framework that integrates prior Physical Wireless Channels into the denoising Diffusion process. The framework employs a dedicated architecture named Fused-Former, which contains a fusion module and a self-attention module that jointly capture the temporal and spectral characteristics of the signals throughout the diffusion trajectory. By leveraging prior wireless channels, PWC-Diff learns to progressively "de-channel" the received signal and recover a representation closer to the transmitted signal. Extensive experiments on several datasets across three WSR tasks have achieved state-of-the-art (SOTA) performance, which demonstrates the rationality of our theory, and ablation experiments further illustrate the effectiveness of our proposed PWC-Diff. Code is available at https://github.com/BUPT-GAMMA/FoundWSR.

## 1. Introduction

Radio signals are ubiquitous in modern communication, serving as the backbone of real-time interactions between

[1]Pengcheng Laboratory, Shenzhen, China [2]Department of Computer Science, Beijing University of Posts and Telecommunications, Beijing, China. Correspondence to: Hui Wang <wangh06@pcl.ac.cn>, Chuan Shi <shichuan@bupt.edu.cn>.

*Proceedings of the 43rd International Conference on Machine Learning*, Seoul, South Korea. PMLR 306, 2026. Copyright 2026 by the author(s).

people and devices (Vinciarelli et al., 2008). With the advancement of next-generation wireless technologies, achieving fast and robust radio signal understanding has emerged as a central objective in sixth-generation (6G) networks (Al-sharif et al., 2020), driving significant research interest in Wireless Signal Recognition (WSR) (Li et al., 2019). WSR leverages artificial intelligence (AI) (Wang et al., 2020b) to automatically infer key signal characteristics, such as modulation type (Meng et al., 2018). The extracted signal attributes provide valuable contextual priors that facilitate more efficient and reliable downstream processing (Elde-merdash et al., 2016). Owing to its broad applicability, WSR has become a critical enabling technology in radio systems, and is widely regarded as a cornerstone task for future 6G networks (Li et al., 2019).

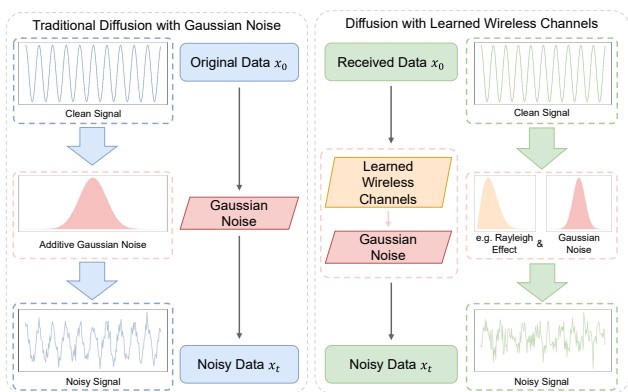

*Figure 1.* The comparison of the forward process between the traditional diffusion models and the proposed method.

Current WSR models, such as IQFormer (Shao et al., 2024) and SpectrumFM (Zhou et al., 2025), have demonstrated strong performance in both single-task and multi-task settings. However, the ultimate goal of WSR is to infer the intrinsic characteristics of the transmitted signal, and in real-world communication systems, received signals are inevitably corrupted by non-ideal channel effects and additive noise. Current models operate solely on the received signal, extracting features and performing recognition without explicitly accounting for the underlying transmission process, which significantly degrades the recognition accuracy (Wang et al., 2020a). To address this limitation, some

researchers may use diffusion models (Yang et al., 2023), which have achieved remarkable success in image denoising by iteratively adding Gaussian noise in the forward process and removing it in the reverse process, showcasing powerful denoising capabilities (Dong et al., 2021). However, in radio signal understanding, channel-induced distortions are typically non-Gaussian (Du et al., 2020). As a result, conventional diffusion frameworks that rely solely on Gaussian noise limit their applicability to practical wireless scenarios.

To effectively learn the characteristics of transmitted signals from channel-distorted observations (Jeruchim et al., 2000), diffusion models should incorporate physical channel effects, not just Gaussian noise. Therefore, we propose **PWC-Diff**, a **Diff**usion model that incorporates prior **P**hysical **W**ireless **C**hannels. As shown in Figure 1, moving beyond denoising toward de-channeling, PWC-Diff superimposes physically plausible channel distortions, whose parameters are estimated from the received signals, together with Gaussian noise to enable joint learning of denoising and de-channeling for improved reconstruction of the transmitted signals. Guided by communication theory, we formulate the diffusion process in the spectral domain, where channel convolution simplifies to element-wise multiplication. For prior channel models, our analysis shows that directly modeling them with a generic neural network (NN) is insufficient. Therefore, we design a lightweight channel parameter estimation network that predicts the parameters of candidate channel models from the received signal, thereby instantiating a physically plausible distortion for the forward process. In the reverse process, to capture both temporal and spectral characteristics, we design FusedFormer, a network that fuses temporal and spectral information through feature fusion and self-attention modules (Hu, 2019), making it well-suited for diffusion-based radio signal understanding. Experiments on multiple WSR tasks show that PWC-Diff achieves state-of-the-art (SOTA) performance, confirming the effectiveness of the framework. Ablation studies further support the validity of our design and theoretical insights.

Our contributions are summarized as follows.

- To the best of our knowledge, we are the first to propose a denoising diffusion framework that integrates prior physical wireless channel models into the forward process for radio signal understanding, emphasizing *de-channeling* as a key principle.

- We theoretically show that directly modeling physical channels with a generic NN is inadequate. To address this, we design a lightweight *channel parameter estimation network* and the FusedFormer, which enable effective de-channeling within a diffusion framework for robust radio signal understanding.

- PWC-Diff achieves SOTA performance against 11

baselines across three WSR tasks, with ablation studies confirming the necessity of our design choices and the advantage of physics-informed diffusion for WSR.

**Conflict of Interest Disclosure**. The authors declare no conflict of interest.

## 2. Preliminary

### 2.1. In-phase and Quadrature (IQ) Data

The received radio signals are typically represented as complex data, i.e., $y = I + jQ$, where $I$ and $Q$ are the in-phase and quadrature components, respectively, and $j = \sqrt{-1}$. However, most deep learning frameworks (e.g., PyTorch) lack efficient support for complex arithmetic. Since real-valued neural networks can implicitly capture the coupling between $I$ and $Q$ with sufficient capacity, we represent an IQ sequence of $L$ as a real matrix $y \in \mathbb{R}^{2 \times L}$, with the first and second rows corresponding to $I$ and $Q$, respectively.

Although IQ data directly encode temporal information, spectral features are not readily apparent. We thus apply the Fourier transform $Y = \mathfrak{F}(y) = \int_{-\infty}^{+\infty} y(t) \cdot \exp(-j2\pi ft)dt$ to obtain the complex spectrum, and we also represent it as a $\mathbb{R}^{2 \times L}$ real matrix using its real and imaginary parts. The original IQ can be recovered by the inverse Fourier transform $y = \mathfrak{F}^{-1}(Y) = \int_{-\infty}^{+\infty} Y(f) \cdot \exp(j2\pi ft)df$.

### 2.2. Communication System and Signal Recognition

In typical communication systems, received signals are distorted by channel impairments and additive noise introduced by the propagation environment (Schmid et al., 2025). Consequently, the received signal can be modeled as

$$y[i] = h[i] * x[i] + n[i], \tag{1}$$

where $i$ denotes the discrete-time sampling index, $h[i]$ captures the channel response such as multipath fading, $*$ represents the linear convolution that is commonly used in communication systems, $x[i]$ is the transmitted signal, and $n[i]$ is additive white Gaussian noise (AWGN). To ensure reasonable system energy efficiency, we adopt the standard normalization assumption from (Tse & Viswanath, 2005):

**Assumption 2.1.** The average power of the channel gain satisfies $\mathbb{E}[|h|^2] \leq 1$.

Under this model, the goal of the WSR task is to learn a discriminative mapping $f_\theta : \mathbb{R}^{2 \times L} \to \mathcal{Y}$ that accurately predicts the semantic label of the transmitted signal $x$ using the observed received signal $y$.

### 2.3. Denoising Diffusion Model

The traditional diffusion model such as DDPM (Ho et al., 2020), adopt a *noise-reconstruction* paradigm to model data

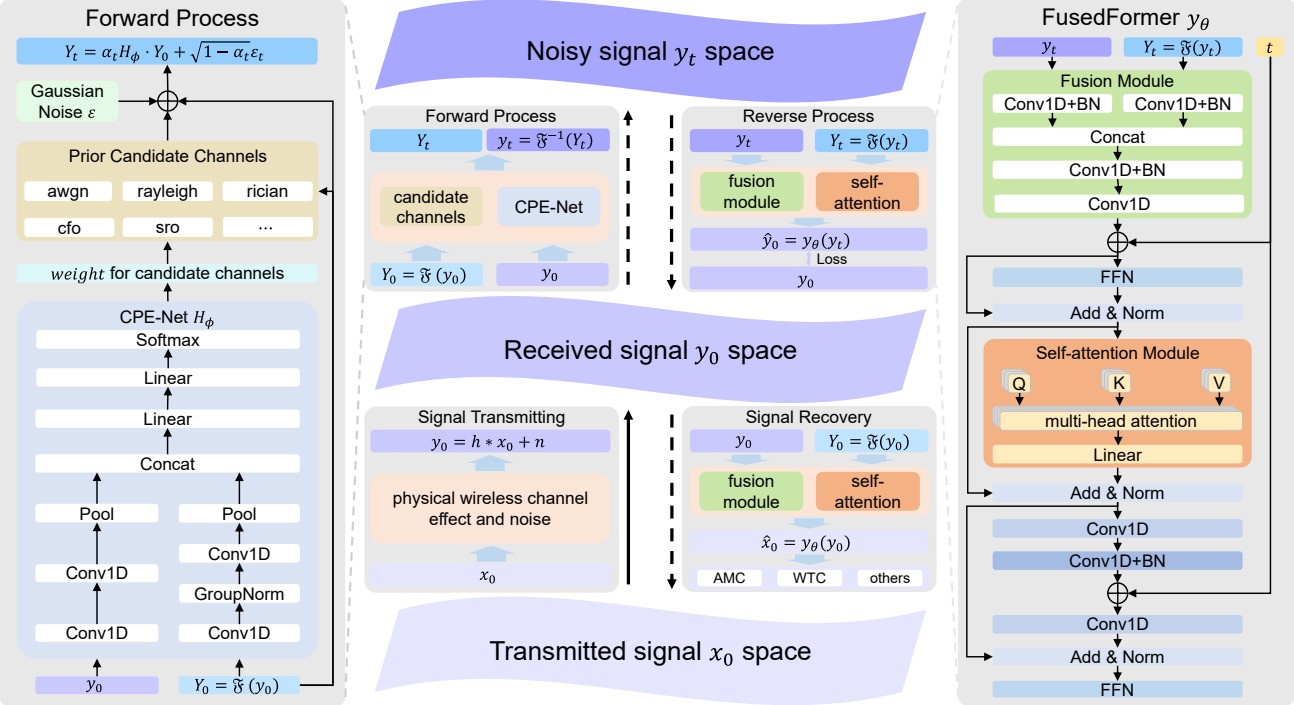

*Figure 2.* The overall architecture of our proposed PWC-Diff. The forward process models physical signal degradation by combining a learned channel perturbation, which apply CPE-Net over predefined candidate channels, with Gaussian noise to obtain $y_t$ from $y_0$. The reverse process, implemented by FusedFormer, reconstructs the received signal $y_0$ by effectively performing de-channeling and denoising. At inference, the trained model processes $y_0$ to extract de-channelized representations for downstream WSR tasks.

distributions. In the forward process, Gaussian noise is incrementally added to the clean data $z_0$ over $T$ discrete timesteps, gradually transforming it into an approximately isotropic Gaussian distribution. This process admits a closed-form expression that directly relates the noisy sample $z_t$ at timestep $t$ to the original data $z_0$:

$$z_t = \alpha_t z_0 + \beta_t \varepsilon, \tag{2}$$

where $\alpha_t^2 + \beta_t^2 = 1$, $\varepsilon \sim \mathcal{N}(0, I)$. In the reverse process, a learnable NN such as U-Net (Ronneberger et al., 2015) parameterized by $\theta$, is trained to estimate the noise or the clean data at each step. The reverse transition is typically modeled as a Gaussian distribution: $p_\theta(z_{t-1}|z_t) = \mathcal{N}(\mu_\theta(z_t, t), \sigma_t^2 I)$. Through reparameterization and simplification, the optimization objective of the model is

$$\mathcal{L} = ||\varepsilon_\theta(z_t, t) - \varepsilon||^2, \tag{3}$$

where $\varepsilon_\theta(z_t, t)$ is a function derived from $z_t$ and $\mu_\theta(z_t, t)$. Although diffusion models are originally developed for generative tasks, they have also proven to be effective for representation learning, such as DDAE (Xiang et al., 2023). Following this paradigm, our work adopts a pre-trained diffusion model and directly utilizes its extracted embeddings for radio signal understanding, enabling accurate inference of radio signal properties.

## 3. Methodology

### 3.1. Overview

We propose PWC-Diff, a diffusion framework that jointly removes channel distortions and noise to recover representations closer to the transmitted signal for WSR. As illustrated in Figure 2, we follow the physical communication system and start from the received signal $y_0$, which is affected by physical wireless channel effect and noise. In the forward process, we progressively corrupt $y_0$ with learned channel effects and Gaussian noise. In the reverse process, a main model learns to invert this degradation, effectively performing de-channeling and denoising. During pretraining, the model learns to map $y_t \rightarrow y_0$. At inference, we assume that the learned CPE-Net approximates the true channel distribution, so we apply the trained main model to $y_0$ and obtain the representation $\hat{x}_0$ for downstream WSR tasks. To capture the rich temporal-spectral characteristics, we follow SpectrumFM and IQformer and design FusedFormer as the main model, which is specifically adapted for diffusion. Details are provided in Section 3.2 and Section 3.3.

### 3.2. Diffusion with Prior Physical Wireless Channels

**Forward Process**. Given IQ data $y_0 \in \mathbb{R}^{2 \times L}$ and the communication model $y = h * x + n$, we formulate the

forward diffusion process by progressively corrupting $y_0$ with physical channel effects and Gaussian noise over $t$ steps. The noisy observation $y_t$ at step $t$ is obtained by

$$y_t = \sqrt{\alpha_t}(y_0 * h) + \sqrt{1 - \alpha_t}\delta_t, \qquad (4)$$

where $*$ denotes convolution, $\delta_t$ is Gaussian noise, and $h$ represents the channel impulse response. Since real-world channels are complex and unknown, we do not model $h$ as a free NN. Instead, we assume $h$ belongs to a set of predefined physical channel models and use a lightweight network to estimate its parameters from $y_0$. This yields a parameterized channel model $h_\phi(y_0)$, leading to the forward process

$$y_t = \sqrt{\alpha_t}(y_0 * h_\phi(y_0)) + \sqrt{1 - \alpha_t}\delta_t. \qquad (5)$$

To reduce computational complexity, we move to the spectral domain via the Fourier transform $Y = \mathfrak{F}(y)$, where convolution becomes element-wise multiplication. Equation 5 is thus reformulated as

$$Y_t = \sqrt{\alpha_t}(H_\phi(Y_0) \odot Y_0) + \sqrt{1 - \alpha_t}\varepsilon_t, \qquad (6)$$

where $\odot$ is the element-wise product operator, and $H_\phi(Y_0) = \mathfrak{F}(h_\phi(y_0))$. Since the Fourier transform of Gaussian noise remains Gaussian, we set $\varepsilon_t \sim \mathcal{N}(0, I)$. As our goal is discriminative WSR, not generation, we do not require the forward process to be strictly monotonic. It suffices that the reverse process can recover a representation close to the original transmitted signal from multiple noisy observations.

**Reverse Process**. The reverse process of the diffusion model aims to recover the original signal from noisy observations. To derive the optimization objective, we express Equation 6 in a probabilistic form as the forward transition distribution, i.e., $q(Y_t|Y_0) = \mathcal{N}(H_\phi(Y_0) \odot Y_0, (1 - \alpha_t)I)$. In the reverse phase, we approximate the posterior distribution using a parameterized Gaussian distribution, i.e., $p_\theta(Y_0|Y_t) = \mathcal{N}(Y_\theta(Y_t, t), \sigma_t^2 I)$, where $\sigma_t$ is the variance, and we do not optimize it explicitly. Our goal is to maximize the log-likelihood $\log p_\theta(Y_0)$. We adopt the Evidence Lower Bound (ELBO) and define the training objective as

$$\begin{aligned} \log p_\theta(Y_0) &= \log \int p_\theta(Y_0, Y_t)dY_t \\ &= \log \int \frac{p_\theta(Y_0, Y_t)}{q(Y_t|Y_0)}q(Y_t|Y_0)dY_t. \end{aligned} \qquad (7)$$

According to Jensen's inequality, we can obtain

$$\begin{aligned} \log p_\theta(Y_0) &\geq \mathbb{E}_{q(Y_t|Y_0)}[\log \frac{p_\theta(Y_0, Y_t)}{q(Y_t|Y_0)}] \\ &= \mathbb{E}_{q(Y_t|Y_0)}[\log p_\theta(Y_0|Y_t)] \\ &\quad - D_{KL}(q(Y_t|Y_0)||p(Y_t)). \end{aligned} \qquad (8)$$

We follow the diffusion-inspired corruption schedule for representation learning, and omit the KL term. Therefore, we optimize the following objective.

$$\mathcal{L} = \mathbb{E}_{Y_0, t, \varepsilon}[\log p_\theta(Y_0|Y_t)]. \qquad (9)$$

Under the Gaussian posterior assumption, the final loss function for model optimization is derived as

$$\mathcal{L} = \mathbb{E}_{Y_0, t, \varepsilon}[||Y_0 - Y_\theta(Y_t, t)||^2]. \qquad (10)$$

We keep this optimization goal according to the work of (Li & He, 2025). Applying the chain rule to $\mathcal{L}$, we obtain the gradient with respect to $\phi$ as

$$\begin{aligned} \nabla_\phi \mathcal{L} = &-2\mathbb{E}[(Y_0 - Y_\theta(Y_t, t))^T \cdot \nabla_{Y_t} Y_\theta(Y_t, t) \\ &\cdot \alpha_t \cdot \nabla_\phi(H_\phi(Y_0) \odot Y_0)], \end{aligned} \qquad (11)$$

which means that $\phi$ and $\theta$ can be optimized jointly.

### 3.3. Neural Network Architecture

In our work, we introduce two NN components: the channel parameter estimation network (CPE-Net) $H_\phi$ and the main denoising model (FusedFormer) $y_\theta$.

**Channel Parameter Estimation Network**. Direct estimation of blind channels from received signals remains theoretically challenging (Wang et al., 2020a), and no suitable foundational model exists for this task. A naive approach using a generic NN tends to converge to the trivial solution $H_\phi \equiv 1$ (see Appendix A), reducing our model to standard DDPM and losing de-channeling capability.

To address this, we adopt a candidate-based strategy. We predefine a set of physical channel models (e.g., Rician fading, multipath) informed by dataset metadata. CPE-Net then learns temporal–spectral features from $Y_0$ through convolutional networks and Multilayer Perceptron (MLP) to predict parameters of these candidate models (e.g., the Rician $K$-factor). This constrained formulation avoids degenerate solutions and provides sufficient expressiveness to validate our framework (demonstrated in Appendix A).

**FusedFormer**. To jointly capture temporal and spectral characteristics, we propose FusedFormer, inspired by IQ-Former. Given the input IQ signal $\boldsymbol{y} \in \mathbb{R}^{B \times 2 \times L}$ and its spectrum $\boldsymbol{Y} = \mathfrak{F}(\boldsymbol{y})$, we first use a 1D convolution layer (denoted as Conv1D) to process the temporal IQ and the spectrum, respectively, which allows us to extract features while maintaining the relationship between the two dimensions in the complex domain. This can be formulated as

$$\begin{aligned} \boldsymbol{y}^{conv} &= \text{Conv1D}(\boldsymbol{y}), \boldsymbol{y} \in \mathbb{R}^{B \times 2 \times L}, \\ \boldsymbol{Y}^{conv} &= \text{Conv1D}(\boldsymbol{Y}), \boldsymbol{Y} \in \mathbb{R}^{B \times 2 \times L}. \end{aligned} \qquad (12)$$

Then, we can apply another Conv1D to traverse and mix the embeddings to obtain the initial representation of the

signals. The formulation can be expressed as

$$\tilde{\boldsymbol{y}} = \text{Conv1D}(\text{BN}(\text{Conv1D}(\text{Concat}(\boldsymbol{y}^{conv}, \boldsymbol{Y}^{conv})))), \tag{13}$$

where $\text{BN}(\cdot)$ represents the batch normalization, $\text{Concat}(\cdot, \cdot)$ represents the concatenation operation. Next, to encode the diffusion step $t$, we simply add these two embeddings together by $\tilde{\boldsymbol{y}} = \tilde{\boldsymbol{y}} + t$. The fused embeddings can be further processed by the self-attention module and residual paths. Specifically, we first introduce a feedforward network (FFN) with two linear layers to pre-process the embeddings with nonlinear GELU activation functions, and then introduce a multi-head self-attention network (denoted as MultiHead) to extract the dependencies between the various dimensions of the representation. We can formalize this as

$$\boldsymbol{y}^{out} = \boldsymbol{y}^f + \text{MultiHead}(\boldsymbol{W}^Q \boldsymbol{y}^f, \boldsymbol{W}^K \boldsymbol{y}^f, \boldsymbol{W}^V \boldsymbol{y}^f), \tag{14}$$

where $\boldsymbol{y}^f = \tilde{\boldsymbol{y}} + \text{FFN}(\tilde{\boldsymbol{y}})$, and $\boldsymbol{W}^Q, \boldsymbol{W}^K, \boldsymbol{W}^V \in \mathbb{R}^{c \times c_h}$, $c_h = c/N_h$ denotes the dimension of each attention head, and $N_h$ is the number of head. Following the design of SpectrumFM, we further enhance the feature map $\boldsymbol{y}^{out}$ and timestep embedding $t$ by first applying a 1D point-wise convolution coupled with a Gated Linear Unit (GLU). Subsequently, a 1D depth-wise convolution with kernel size 3 captures the local embedding structure, and a final 1D convolution projects the output into compact local representations. Formally,

$$\boldsymbol{y}^{\text{local}} = \boldsymbol{y}^{\text{out}} + \text{Conv1D}(\text{BN}(\boldsymbol{z}_2)),$$
$$\text{where} \quad \boldsymbol{z}_2 = \text{Conv1D}(\text{GLU}(\boldsymbol{z}_1); \text{kernel} = 3), \tag{15}$$
$$\boldsymbol{z}_1 = \text{Conv1D}(\boldsymbol{y}^{\text{out}}) + t.$$

Finally, we use another FFN to combine the results, i.e., $\boldsymbol{y}^{\text{final}} = \boldsymbol{y}^{\text{local}} + \text{FFN}(\boldsymbol{y}^{\text{local}})$. The encoder can stack multiple layers to obtain the final representation.

During the pretraining stage, we apply a simple linear decoder to $\boldsymbol{y}^{\text{final}}$ and align it to the original data $\boldsymbol{y}_0$, and optimize the NN following Equation 10. During the fine-tuning stage, we use global average pooling to $\boldsymbol{y}^{\text{final}}$ and feed it to a classifier for the WSR task. The entire network is fine-tuned with standard cross-entropy loss.

## 4. Experiments

In this section, we carry out a series of experiments on several datasets and WSR tasks to illustrate three key aspects. **Q1**: The performance gain of PWC-Diff over established baselines. **Q2**: The effectiveness of the proposed pretraining strategy. **Q3**: The necessity and impact of incorporating suitable candidate channel models.

### 4.1. Experimental Settings

The PWC-Diff is implemented with PyTorch (Paszke et al., 2019) and optimized with Optuna (Akiba et al., 2019). The

implementation details and the use of hyper-parameters are listed in Appendix C and Appendix D.

**Datasets**. We use four datasets across three WSR tasks in our experiments. The datasets are RML2016.10A (O'shea & West, 2016), RML2022 (Sathyanarayanan et al., 2023) for Automatic Modulation Classification (AMC), TechRec (Fontaine & Shahid, 2023) for Wireless Technology Classification (WTC), and GNSS (Ghanbarzade & Soleimani, 2025) for Anomaly Detection (AD). RML2016.10A, RML2022, and TechRec contain several property types (modulation or wireless technology type) under various Signal-to-Noise Ratio (SNR) conditions, while GNSS contains 6 types of signals under a single SNR condition. For RML2016.10A, RML2022, and TechRec, we follow the setting of IQFormer, which uses 80% of the data for training and 20% for testing, while GNSS provides its own test set. The evaluation metric is the Top-1 accuracy. The dataset and task details are presented in Appendix B.

**Baselines**. We implement 11 baseline models for our main experiments. The baselines are AMC_Net (Zhang et al., 2023), CGDNN (Njoku et al., 2021), CNN2 (O'Shea et al., 2018), GRU2 (Hong et al., 2017), MCNet (Huynh-The et al., 2020), MSNet (Zhang et al., 2021), ResNet (Liu et al., 2017), Transformer (Vaswani et al., 2017), VGG (O'Shea et al., 2018), IQFormer (Shao et al., 2024), SpectrumFM (Zhou et al., 2025), where the first 10 models are trained using standard supervised training, while for SpectrumFM, we perform pre-training and fine-tuning on the selected dataset following its original pipeline.

*Table 1.* The channel model set and candidate channel models for each dataset.

| Dataset | Channel Model |
|---|---|
| RML2016.10A | rician, cfo, awgn, sro |
| RML2022 | rician, cfo, multipath, sro, awgn |
| TechRec | awgn |
| GNSS | rician, cfo, awgn, sro |
| Used Channel Models | rician, rayleigh, nakagami, cfo, sro, awgn, multipath |

**Candidate Channel Models**. Since we assume that the distribution of the noisy channel model is consistent with the distribution of the data channel, we select different candidate channels based on the survey results and the dataset processing method. The set of channel models and the candidate channel model for each dataset is listed in Table 1, where "rician", "rayleigh", and "nakagami" denote common small-scale fading models; "cfo" and "sro" refer to carrier frequency offset and sampling rate offset, respectively; and

*Table 2.* The overall accuracy of different models (percentage), where the "Average" represents the average performance and the "Best" represents the best performance among all SNRs on 4 datasets. The last column represents the average rank of each model (The best performance is represented in bold, and the second-best performance is underlined).

| Model | RML2016.10A | | RML2022 | | TechRec | | GNSS | Rank |
|---|---|---|---|---|---|---|---|---|
| | Average | Best | Average | Best | Average | Best | Average | |
| AMC_Net | 57.92 | 84.68 | 68.48 | 98.05 | 86.62 | 98.52 | 83.19 | 4.4 |
| CGDNN | 47.38 | 71.18 | 59.28 | 85.45 | 57.65 | 76.41 | 63.08 | 10.9 |
| CNN2 | 48.64 | 69.91 | 62.99 | 89.30 | 68.64 | 80.98 | 84.23 | 8.7 |
| GRU2 | 60.86 | 90.36 | 67.15 | 97.98 | 37.79 | 39.09 | 78.38 | 6.9 |
| MCNet | 54.01 | 84.95 | 66.19 | 98.32 | 69.23 | 83.51 | 82.12 | 6.7 |
| MSNet | 60.02 | 89.59 | 65.50 | 91.64 | 88.79 | 97.65 | 78.26 | 5.4 |
| ResNet | 50.25 | 80.91 | 36.18 | 59.36 | 70.36 | 78.63 | 61.66 | 10.1 |
| Transformer | 57.40 | 85.95 | 67.38 | 97.20 | 70.36 | 78.63 | 76.52 | 7.3 |
| VGG | 50.69 | 79.09 | 56.96 | 82.32 | 77.20 | 84.44 | 82.23 | 8.3 |
| IQFormer | 63.54 | 93.59 | 69.22 | 98.61 | 85.00 | 90.30 | 78.26 | 3.4 |
| SpectrumFM | 59.85 | 90.14 | 68.87 | 97.82 | 79.63 | 87.10 | 83.64 | 4.4 |
| PWC-Diff | **63.60** | **93.91** | **69.97** | **98.64** | **91.02** | **99.23** | **84.75** | **1.0** |

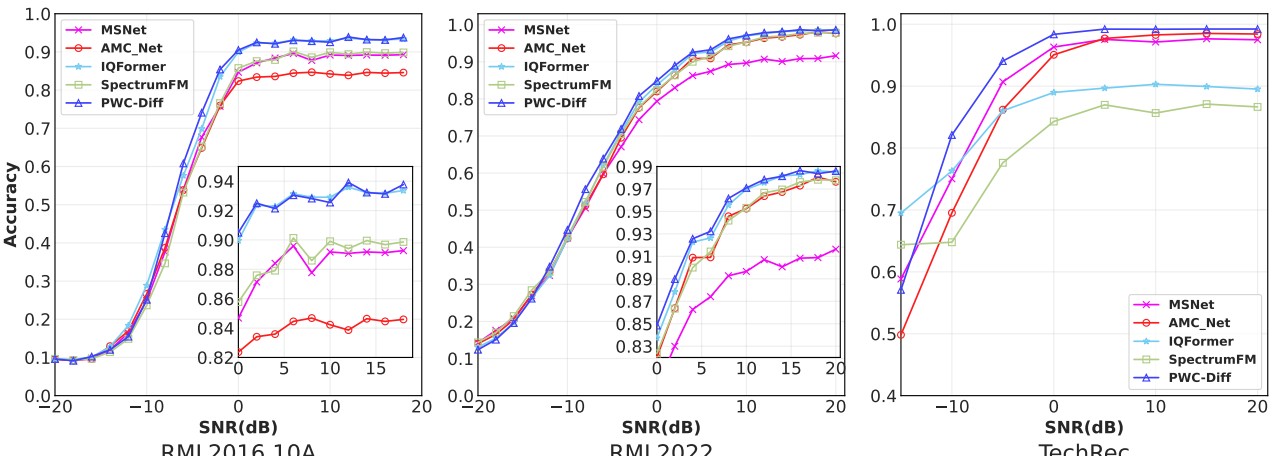

*Figure 3.* Visualization of performance under different SNR conditions (left: RML2016.10A, center: RML2022, right: TechRec).

"awgn" stands for additive white Gaussian noise. The term "multipath" indicates the presence of frequency-selective fading due to multiple propagation paths.

### 4.2. Main Results (Q1)

In this section, we provide a comparison between the PWC-Diff and the 11 baselines. The performance is listed in Table 2. Additional experiments are presented in Appendix E.

**Overall Performance**. From the results, we find that the PWC-Diff achieves the best performance among the four datasets across three tasks, with about 0.89% improvement on average compared to the best baseline. This consistent improvement validates the effectiveness of our physics-informed pretraining strategy and the FusedFormer archi-

tecture. Compared with IQFormer, the SOTA AMC model, PWC-Diff achieves about 3.33% improvement across multiple WSR scenarios. This gain is particularly pronounced in challenging environments, such as RML2022. These results demonstrate that explicitly modeling physical channel priors in the diffusion process leads to more robust and generalizable signal representations.

**Performance under Various SNR Conditions**. To further demonstrate the performance of PWC-Diff under various SNR conditions, we compare it with AMC_Net, MSNet, IQFormer, and SpectrumFM for a clearer illustration. Figure 3 shows their classification accuracy on RML2016.10a, RML2022, and TechRec. At extremely low SNRs (-20dB), all models fail as the signal is fully dominated by noise. As SNR increases, performance gradually improves across all

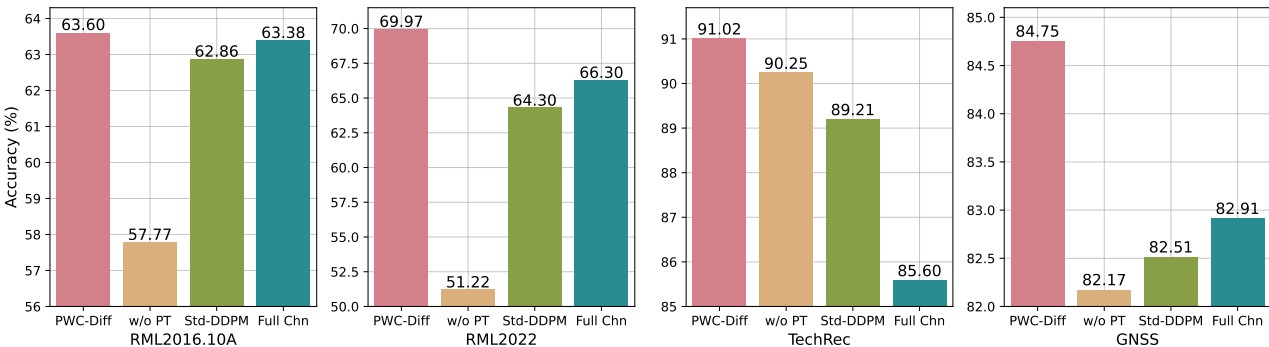

*Figure 4.* Ablation experimental results regarding the PWC-Diff pre-training strategy (The dataset from left to right is RML2016.10A, RML2022, TechRec, and GNSS).

*Table 3.* The ablation results of different candidate channel models (percentage), where we present the average and the best accuracy among different SNRs on RML2016.10A.

|  | PWC-Diff | w/o cfo | w/o rician | w/o awgn | w/o sro | w/ rayleigh | w/ multipath | w/ nakagami |
|---|---|---|---|---|---|---|---|---|
| Average | **63.60** | 61.46 | 62.28 | 61.74 | 61.38 | 61.77 | 63.21 | 62.53 |
| Best | **93.91** | 91.32 | 92.82 | 91.91 | 91.64 | 91.86 | 93.77 | 92.68 |

*Table 4.* The ablation results of different candidate channel models (percentage), where we present the average and the best accuracy among different SNRs on RML2022.

|  | PWC-Diff | w/o cfo | w/o rician | w/o awgn | w/o sro | w/o multipath | w/ rayleigh | w/ nakagami |
|---|---|---|---|---|---|---|---|---|
| Average | **69.97** | 68.50 | 68.38 | 68.52 | 68.45 | 68.69 | 69.09 | 68.75 |
| Best | **98.64** | 97.16 | 97.18 | 97.57 | 97.43 | 97.32 | 97.50 | 97.36 |

methods. Notably, PWC-Diff consistently outperforms competitors, maintaining strong accuracy in low-SNsR regimes (-10 dB to 0 dB) and achieving the highest performance at higher SNRs (> 0 dB). This demonstrates its superior ability to recover signal-intrinsic features despite channel distortions and noise, highlighting its effectiveness for radio signal understanding.

### 4.3. Ablation Study

We conduct a series of ablation studies to validate our two claims: it is important to integrate physical channel effects into the pretraining strategy of the diffusion model (**Q2**), and reasonable channel estimation of the radio signal data is essential (**Q3**).

**The Importance of PWC-Diff's Pretraining Strategy**. To validate this, we evaluate three variants: (1) directly fine-tuning FusedFormer without pretraining (*w/o PT*), (2) using a standard diffusion model with Gaussian noise only (*Std-DDPM*), and (3) adding channel effects during fine-tuning (*Full Chn*). The results are shown in Figure 4. Compared to *w/o PT*, PWC-Diff consistently outperforms the model FusedFormer on both "Average" and "Best" among the four

datasets, confirming that the performance gain primarily stems from our pretraining strategy. The comparison with *Std-DDPM* further supports this: by incorporating physical channel priors during pretraining, FusedFormer learns more robust representations for signal property recognition. Notably, *Full Chn* yields representations of the received signal, whereas PWC-Diff recovers features closer to the transmitted signal. The superior performance of PWC-Diff over *Full Chn* validates our de-channelized learning objective and demonstrates that removing channel effects during representation learning significantly benefits WSR. The above results are able to demonstrate the importance of our proposed pretraining strategy.

**The Essential of a Reasonable Channel Estimation**. We conduct two sets of ablation studies to validate the design of our candidate channel set. First, on RML2016.10a, we respectively remove "cfo", "rician", "awgn", and "sro" from the candidate set, and separately add "rayleigh", "multipath", and "nakagami". Similarly, on RML2022, we remove each existing candidate and add "rayleigh" or "nakagami". The results are reported in Table 3 and Table 4, where "w/o" denotes removal and "w/" denotes addition of a channel model. We can find that PWC-Diff consistently outperforms

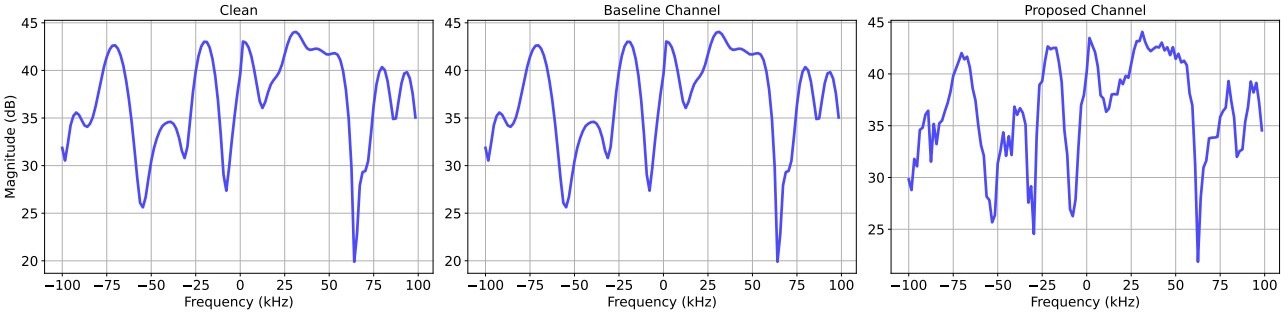

*Figure 5.* Comparison of signal spectrum under different channel modeling strategies (The sample is QAM64 modulation under 16dB). Clean is the original data, baseline channel denotes learning channel with simple NN, and proposed channel represents the CPE-Net.

all variants, demonstrating that our selected candidate models are well-suited to the datasets. More importantly, it confirms that reasonable channel modeling is critical for learning transmitted-signal representations. Inappropriate channel assumptions severely degrade both representation quality and downstream recognition performance.

Then, we present a case study to further illustrate that the estimation of the channel model $H_\phi$ cannot be realized by a simple and unconstrained NN. As shown in Figure 5, if we use a simple MLP network equipped with softmax output activation (designed to satisfy the assumption 2.1), the result of $H_\phi$ easily converges to 1. This renders the channel-distorted signal nearly indistinguishable from the original, ultimately collapsing to standard Gaussian denoising. In contrast, our CPE-Net produces meaningful and non-degenerate channel estimates, leading to visibly distinct perturbations. This highlights the necessity of incorporating structural priors into channel modeling. Future work may explore more principled forms of blind channel adaptation, building upon this constrained estimation framework.

## 5. Related Work

### 5.1. Wireless Signal Recognition Models

Early WSR approaches rely on handcrafted features and manual annotation, which are labor-intensive and limit scalability (Zhao et al., 2017; Ahmad et al., 2010). The advent of deep learning has significantly advanced the field (Zhang et al., 2025), with CNN- and Transformer-based architectures such as AMC_Net (Zhang et al., 2023) and IQ-Former (Shao et al., 2024), which achieve SOTA performance in AMC and other WSR tasks. More recently, self-supervised representation learning has gained traction, leveraging large-scale unlabeled data to learn general-purpose signal representations. Strategies include contrastive learning (Hu et al., 2024) and reconstruction-based pretraining (Milosheski et al., 2025). Notably, SpectrumFM (Zhou et al., 2025) adopts a BERT-style "mask-reconstruction"

paradigm (Devlin et al., 2019) to build a foundational model for various WSR tasks. However, these methods operate solely on the received signal, whose representation is inevitably distorted by channel effects and noise. In contrast, our approach explicitly aims to recover the transmitted signal by modeling and reversing physical channel distortions. The resulting de-channelized features are more aligned with the intrinsic signal properties, leading to improved robustness and performance in downstream WSR tasks.

### 5.2. Diffusion Models

Diffusion models have advanced rapidly over the past five years, becoming a dominant paradigm in generative modeling, with widespread success in high-resolution and conditional image generation (Yang et al., 2023). Pioneering works such as DDPM (Ho et al., 2020) establish the foundational denoising framework, while DDIM (Song et al., 2021) extends its solution space through non-Markovian sampling. LDM (Rombach et al., 2022) further enables scalable high-resolution generation by operating in a learned latent space. Moreover, researchers have extended the application of diffusion models to other domains. For example, PeFGL (Sun et al., 2026) employs diffusion models to generate augmented graphs. RF-Diffusion (Chi et al., 2024) applies the time–frequency diffusion model to conditionally generate high-quality radio signals.

Although these methods focus primarily on generation, diffusion models have also shown promise in recognition tasks. For instance, DDAE (Xiang et al., 2023) adopts a reconstruction-based pretraining strategy by training the model to denoise noisy inputs and has achieved remarkable performance in image classification. In the radio signal understanding domain, TS-DDAE (Liu et al., 2026) applies the temporal-spectral diffusion model for radio signal pretraining. Although superficially similar, our approach differs fundamentally. PWC-Diff explicitly incorporates physical wireless channel models into the diffusion process. This enables the model to learn de-channeling capabilities, leading

to more accurate recovery of transmitted-signal characteristics for WSR tasks.

## 6. Conclusion

In this paper, we move beyond denoising toward de-channeling by proposing PWC-Diff, a novel diffusion framework that incorporates prior physical channel models to recover representations closer to the transmitted signal for robust radio signal understanding. PWC-Diff leverages a lightweight CPE-Net to infer instance-specific parameters from received signals, and uses the FusedFormer architecture to jointly exploit temporal and spectral features of radio signals. Across several datasets and WSR tasks, PWC-Diff have achieved SOTA performance. Ablation studies further demonstrate the necessity of each component. Future work will explore tighter integration of channel priors and investigate whether other blind channel adaptation can further enhance the framework.

## Acknowledgments

This work is supported in part by the National Natural Science Foundation of China (No. 62550138, 62192784, 62572064, 62472329), BUPT Excellent Ph.D. Students Foundation (No. CX20241010), the Major Key Project of Peng Cheng Laboratory.

## Impact Statement

This paper presents work whose goal is to advance the field of Machine Learning. There are many potential societal consequences of our work, none of which we feel must be specifically highlighted here. This research strictly adheres to data usage regulations; all experiments are based on public datasets, with the commitment not to process any private information. And current work does not contain ethical topics. However, WSR can support beneficial applications (e.g., spectrum monitoring, anomaly detection) but may also enable surveillance or signal intelligence. Deployment should comply with local laws and be restricted in contexts that infringe on privacy or civil liberties.

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

# A. Additional Derivation

In this section, we extend the derivation in Section 3.2 to show that $H_\phi$ easily converges to 1.

**Proposition A.1.** *Based on the diffusion model using Equation 6 and Equation 10 and Assumption 2.1, without explicit supervision or regularization constraints on $H_\phi$, the loss function $\mathcal{L}$ reaches a local minimum at $H_\phi \equiv \mathbf{1}$, i.e., there exists $\delta > 0$ such that for all $\mathcal{H}$ satisfying $||\mathcal{H} - \mathbf{1}||_\infty < \delta$, we have*

$$\mathcal{L}(\mathcal{H}, \theta^*(\mathcal{H})) \geq \mathcal{L}(\mathbf{1}, \theta^*(\mathbf{1})),$$

*where $\theta^*(\mathcal{H}) = arg\, min_\theta \mathcal{L}(\mathcal{H}, \theta)$.*

*Proof.* We suppose $Y_\theta(Y_t, t)$ is sufficiently learned to approximate any measurable function. For fixed $\mathcal{H}$, the parameter that minimizes $\mathcal{L}$ should satisfy

$$\mathcal{L}^*(H) = \min_\theta \mathcal{L}(\mathcal{H}, \theta) = \mathbb{E}[Var(Y_0|Y_t)].$$

Considering the scalar case, let $y_t = \alpha h y_0 + \beta \varepsilon$, where $\beta = \sqrt{1 - \alpha^2}$, and $h \in (0, 1)$, and define

$$\ell(h) = \mathbb{E}[(y_0 - \mathbb{E}[y_0|y_t])^2].$$

The $\ell(h)$ is monotonically decreasing in $h \in (0, 1)$. Now we present the proof. Let $s = \alpha h > 0$, then $y_t = s y_0 + \beta \varepsilon$. We consider $0 < s_1 < s_2$ corresponding to $h_1 < h_2$. With the same $y_0$ and $\varepsilon$, we define

$$y_t^{(1)} = s_1 y_0 + \beta \varepsilon, y_t^{(2)} = s_2 y_0 + \beta \varepsilon.$$

Now, we can construct a new observation:

$$\tilde{y} = \frac{s_1}{s_2} y_t^{(2)} = s_1 y_0 + \beta \frac{s_1}{s_2} \varepsilon.$$

$\tilde{y}$ and $y_t^{(1)}$ have the same term $s_1 y_0$, while the variance is

$$Var(\beta \frac{s_1}{s_2} \varepsilon) = \beta^2 (\frac{s_1}{s_2})^2 < \beta^2 = Var(\beta \varepsilon)$$

Therefore, they satisfy the following relationship:

$$\mathbb{E}[(y_0 - \mathbb{E}[y_0|\tilde{y}])^2] \leq \mathbb{E}[(y_0 - \mathbb{E}[y_0|y_t^{(1)}])^2].$$

However, $\tilde{y}$ is constructed from $y_t^{(2)}$, so

$$\mathbb{E}[y_0|\tilde{y}] = \mathbb{E}[\mathbb{E}[y_0|y_t^{(2)}]|\tilde{y}].$$

Based on the tower property of conditional expectation, and further by conditional variance decomposition, we can get

$$\mathbb{E}[(y_0 - \mathbb{E}[y_0|\tilde{y}])^2] = \mathbb{E}[(y_0 - \mathbb{E}[y_0|y_t^{(2)}])^2] + \mathbb{E}[(\mathbb{E}[y_0|y_t^{(2)}] - \mathbb{E}[y_0|\tilde{y}])^2] \geq \ell(h_2).$$

Therefore, we obtain

$$\ell(h_2) \leq \mathbb{E}[(y_0 - \mathbb{E}[y_0|\tilde{y}])^2] \leq \ell(h_1),$$

which demonstrate the $\ell(h)$ is monotonically decreasing in $h \in (0, 1)$.

As $\ell$ is continuous at $h = 1$, there exists $\delta > 0$ such that for all $h \in (1 - \delta, 1]$, $\ell(h) \geq \ell(1)$, with equality iff $h = 1$. Then, for any $\mathcal{H}$ that satisfies $||\mathcal{H} - \mathbf{1}||_\infty < \delta$, each component $\mathcal{H}_i(Y_0) \in (1 - \delta, 1]$. So, we have for each $i$,

$$Var(Y_{0,i}|Y_t; \mathcal{H}) \geq Var(Y_{0,i}|Y_t; \mathbf{1}).$$

Taking expectation over $Y_0$ and summing over $i$, we have

$$\mathcal{L}(\mathcal{H}, \theta^*(\mathcal{H})) \geq \mathcal{L}(\mathbf{1}, \theta^*(\mathbf{1})).$$

This completes the proof. □

Then, we demonstrate that our strategy using CPE-Net prevents $H_\phi$ from converging to 1.

**Proposition A.2.** *Based on the diffusion model using Equation 6 and Equation 10 and Assumption 2.1, assume that the channel function $H_\phi : \mathbb{R}^L \to (0, 1]^L$ is constrained to the form*

$$H_\phi(Y_0) = \sum_{k=1}^{K} w_k(Y_0; \phi) h^{(k)},$$

*where:*
*1. $H = \{h^{(k)}\}_{k=1}^{K} \subset (0, 1]^L$ are physical wireless channel models and $\mathbf{1} \notin H$.*
*2. $w_k(Y_0; \phi) = \frac{\exp(s_k(Y_0; \phi))}{\sum_{j=1}^{K} \exp(s_j(Y_0; \phi))} \in (0, 1)$ so that $\sum_k w_k = 1$.*

*Then, for any $\phi \in \Phi$, the minimizer $(\phi^*, \theta^*)$ of $\mathcal{L}(\phi, \theta)$ satisfies*

$$H_{\phi^*} \not\equiv 1.$$

*Proof.* For any $Y_0$ and $\phi$,

$$H_\phi(Y_0) = \sum_{k=1}^{K} w_k(Y_0; \phi) h^{(k)} \in H.$$

Assume, in contradiction, that there exists some $\phi_0$ such that for all $Y_0$, $H_{\phi_0}(Y_0) = 1$. Since $H_{\phi_0}(Y_0) \in H$ for all $Y_0$, this would imply $\mathbf{1} \in H$. However, we require $\mathbf{1} \notin H$. This contradiction shows that no such $\phi_0$. Therefore, the function $H \equiv 1$ is not contained in $\{H_\phi : \phi \in \Phi\}$.

Then, we can define a metric:

$$d = \inf_{h \in H} ||h - \mathbf{1}||_\infty.$$

$d$ is strictly positive as $\mathbf{1} \notin H$. Therefore, for every $\phi \in \Phi$ and all $Y_0$,

$$||\mathcal{H} - \mathbf{1}||_\infty \geq d > 0.$$

Therefore, the $H \equiv 1$ lies outside of the closure of $\{H_\phi : \phi \in \Phi\}$, the minimizer $(\phi^*, \theta^*)$ of $\mathcal{L}(\phi, \theta)$ satisfies

$$H_{\phi^*} \not\equiv 1.$$

This completes the proof. $\square$

# B. Details of Datasets and Tasks

## B.1. Datasets

In this section, we present details of the datasets that we use in this paper. Statistics are listed in Table 5, where SNR denotes the signal-to-noise ratio for RML2016.10A, RML2022, TechRec. The RML2016.10A and RML2022 are the most typical benchmark for AMC task. The TechRec contains LTE, WiFi and DVB-T three types of wireless technology. Directly processing the raw IQ of the TechRec is hard as the length of the signals will cause heavy memory pressure, so we slice them into segments of length 1024 and label them according to the wireless type of signal to which each segment belongs. Moreover, we add Gaussian noise to the TechRec to simulate the SNR following the SpectrumFM, whose SNR ranges from -15dB to 20dB. The GNSS is a GPS jamming signal classification dataset, which contains DME, narrowband, single AM, single chirp, single FM jamming signals and no jamming signal, and we also follow the TechRec to slice raw data into 1024-segments. The GNSS does not contain SNR information, and we retain its original settings to test our model's ability to analyze this type of received data. The RML2016.10A, RML2022, and TechRec take 80% of the data for training and 20% for testing, respectively, while the GNSS provides its own test set, with about 90,000 for training and 22,500 for testing.

*Table 5.* The statistics of the datasets used in this work.

| Dataset | Number of samples | Length of each sample | Number of classes | Min SNR (dB) | Max SNR (dB) | SNR interval (dB) |
|---|---|---|---|---|---|---|
| RML2016.10A | 220,000 | 128 | 11 | -20 | 18 | 2 |
| RML2022 | 462,000 | 128 | 11 | -20 | 20 | 2 |
| TechRec | 202,762 | 1,024 | 3 | -15 | 20 | 5 |
| GNSS | 112,500 | 1,024 | 6 | - | - | - |

### B.2. Tasks

We evaluate our model with three tasks including the AMC, WTC and AD. Here we give descriptions of each task.
**Automatic Modulation Classification (AMC)** (Zhang et al., 2023). In wireless communication, modulation is the process of mapping digital information (bit stream) onto an analog carrier signal in order to transmit it efficiently in the channel. The AMC task aims to identify the modulation type of the received signals.
**Wireless Technology Classification (WTC)** (Bitar et al., 2017). Wireless technology type is a type of wireless communication technology standard or system that has a specific physical layer waveform structure, modulation method, frame format, multiple access mechanism, bandwidth characteristics, and protocol specifications. The WTC task aims to recognize the wireless technology type given the received signals.
**Anomaly Detection (AD)** (Yan et al., 2023). Real-world signals are often mixed with various anomalous signals, such as interference signals. The task of AD is to identify the type of anomalous signal based on the received signal.

## C. Other Implementation Details

In this section, we supplement the details regarding the implementation of our model. First, we provide details about the CPE-Net. Given the input IQ signal $y \in \mathbb{R}^{B \times 2 \times L}$ and and its spectrum $Y = \mathfrak{F}(y)$, we use the Conv1D and Pool layers to extract the features. The outputs are concatenated and then processed by MLP with linear layers and a softmax to obtain the parameters of the candidate channel models. The description of the FusedFormer is presented in Section 3.3. Finally, we describe the classifier architecture used during fine-tuning. The output representation, which has shape $\mathbb{R}^{B \times d \times L}$ with $d$ denoting the hidden dimension, is first aggregated across the temporal dimension via a pooling layer to synthesize per-sample features, and then fed into an MLP to produce the final classification logits.

Then, we give details about the training configuration. The pretraining initialization method of CPE-Net and FusedFormer is the default PyTorch initialization method, i.e., a uniform distribution bounded by 1/sqrt(in_features). During pretraining, the optimizer is "AdamW", with parameters: lr=1e-3, weight_decay=0.01. Our pretraining learning rate strategy is "ReduceLROnPlateau" according to the pretraining loss, with parameters: mode="min", factor=0.5, patience=200, min_lr=1e-12. Moreover, we take an early stop mechanism with patience=500 according to the loss. During the fine-tuning, we modify some settings. The optimizer is the same. The learning rate strategy is "ReduceLROnPlateau" according to the valid set loss, with parameters: mode="min", factor=0.5, patience=3, min_lr=1e-8. The patience of the early stop mechanism is 15 according to the valid set loss. The final model for testing is the one with the least valid loss. The whole network is pretrained and fine-tuned using Tesla A100.

## D. Usage of Hyper-parameters in Experiments

The hyper-parameters we use in our experiments are listed in Table 6, where we list the parameter name, the description of each parameter, and the value. We use the same hyper-parameters for all of the experiments related to PWC-Diff.

## E. Additional Experiments

### E.1. Efficiency Comparison

In this section, we present an efficiency analysis of PWC-Diff. To ensure a fair comparison, we only compare PWC-Diff with SpectrumFM, both of which are pretraining models for WSR. The evaluation dataset is the RML2016.10A with input length 128. The results are shown in Table 7. From the results, we can find that with a similar number of parameters,

*Table 6.* The hyper-parameters used in our experiments.

| Parameter name | Description | Value |
|---|---|---|
| num_layers | The number of FusedFormer layers | 4 |
| max_step | The maximal diffusion steps | 1000 |
| timestep | The step used in fine-tuning | 30 |
| min_noise | The minimal noise added to signals | 4.28e-7 |
| max_noise | The maximal noise added to signals | 0.01189 |
| hidden_dim | The hidden dimension used in the FusedFormer | 256 |
| num_heads | The number of heads used in the attention layer in FusedFormer | 16 |

*Table 7.* The efficiency comparison between PWC-Diff and SpectrumFM on RML2016.10A.

| Model | Parameters | Pretraining time (second) | Inference time (second) |
|---|---|---|---|
| SpectrumFM | 7.98M | 4,379 | 0.606 |
| PWC-Diff | 6.43M | 553 | 0.608 |

PWC-Diff not only achieves superior performance but also has a similar inference time cost to SpectrumFM. Furthermore, under the same patience settings, our method requires significantly less pretraining time, demonstrating the efficiency of our model training and the lower inference overhead. In practice, we also use the *torch.compile* to further optimize our model, and the inference can be faster.

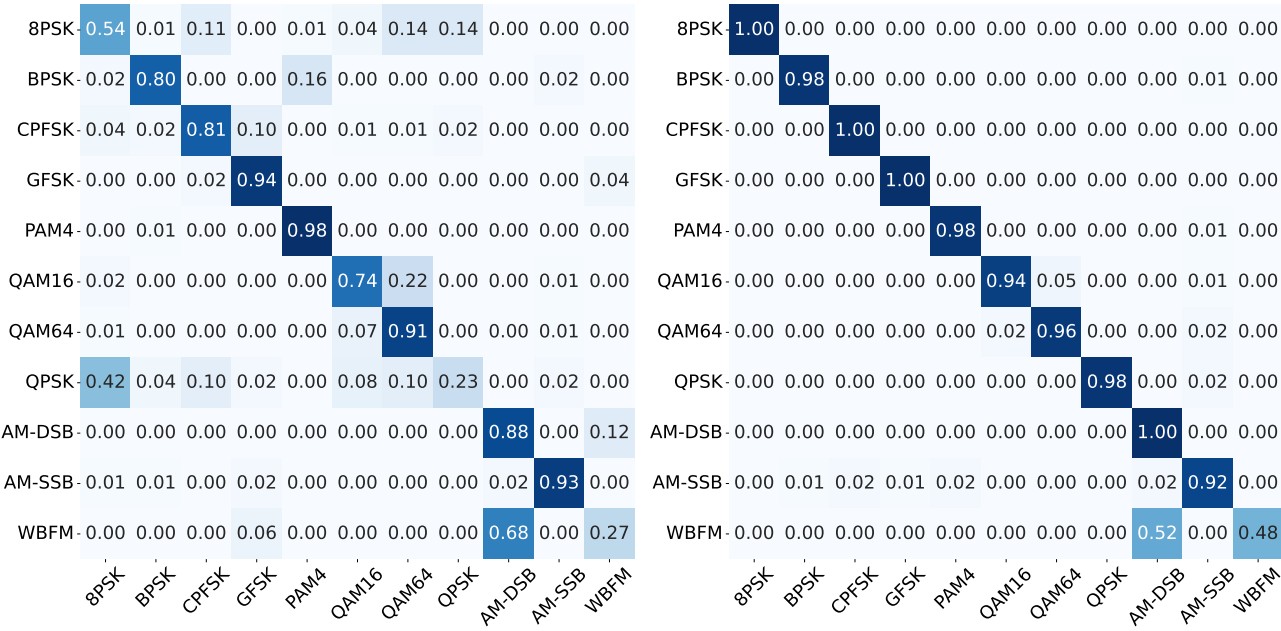

*Figure 6.* Normalized confusion matrix on RML2016.10A dataset (left: -4dB, right: 6dB).

### E.2. Visualization of the PWC-Diff Output

In this section, we give the visualization of the output of the PWC-Diff in the AMC task. The results of the figures are consistent with the conclusions of the SpectrumFM and the IQFormer, and can further verify the the correctness and effectiveness of the PWC-Diff.

First, we give the normalized confusion matrix of PWC-Diff results on the RML2016.10A and the RML2022 dataset with

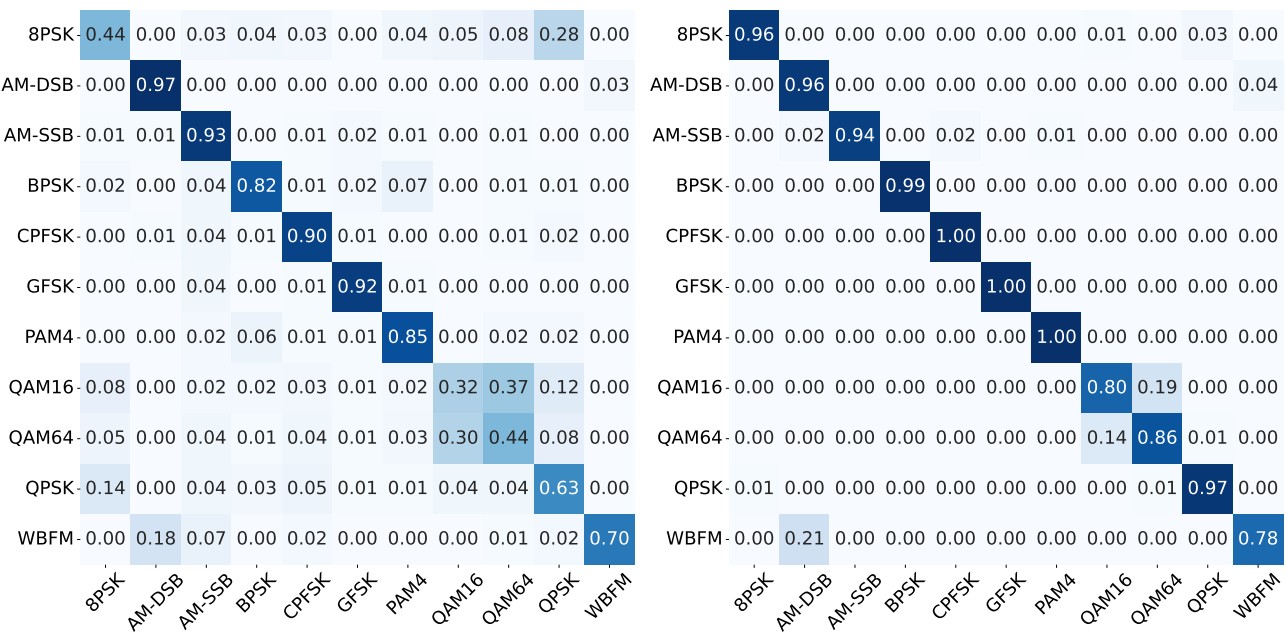

*Figure 7.* Normalized confusion matrix on RML2022 dataset (left: -4dB, right: 6dB).

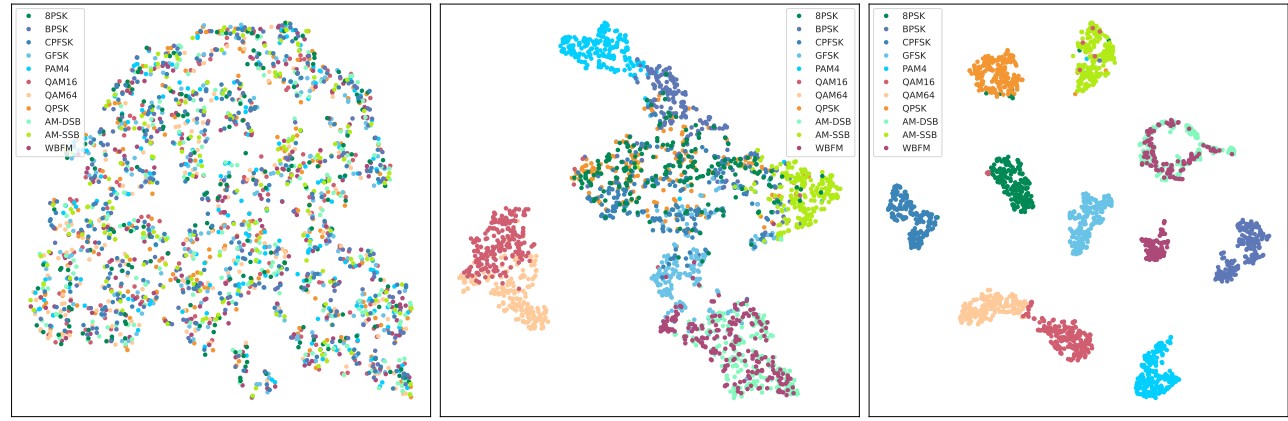

*Figure 8.* Visualization of feature output with t-SNE on RML2016.10A dataset (left: -20dB, center: -6dB, right: 8dB).

-4dB and 6dB SNR, which is shown in Figure 6 and Figure 7. From the results, we can find that under low SNR conditions, noise will seriously affect the PWC-Diff's distinction between the three modulation types: 8PSK, QPSK and WBFM. When the SNR reaches 6dB, the model can already distinguish most modulation types well, even distinguishing some modulation types with 100% accuracy. However, the model can hardly distinguish the WBFM modulation type from the AM-DSB, which is also consistent with the conclusions of the SpectrumFM and the IQFormer.

Then, we present the t-SNE visualizations of the RML2016.10A and the RML2022 dataset under -20dB, -6dB, and 8dB for PWC-Diff. The results are shown in Figure 8 and Figure 9. At a -20dB SNR, the data is filled with noise, leading the model to classify the data randomly. Consequently, the various modulation categories appear mixed together in the t-SNE figure. At a low SNR (-6dB), our PWC-Diff already demonstrates a certain degree of discrimination, with many modulation categories clearly distinguished. At a high SNR (8dB), the model's discrimination is much better. Furthermore, at 8dB, the model fails to distinguish WSFM and AM-DSB modulation categories particularly well, which is consistent with the conclusions in Figure 6 and Figure 7.

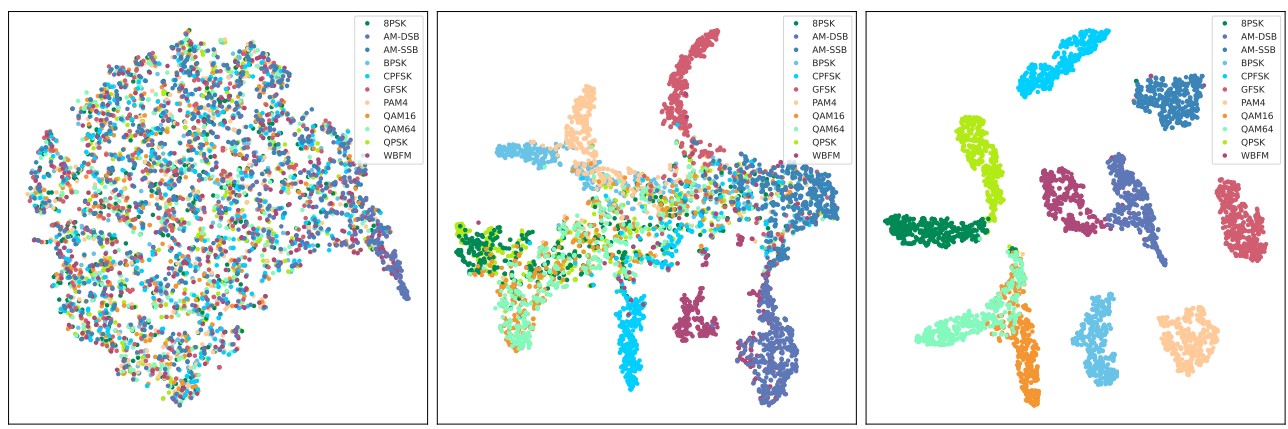

*Figure 9.* Visualization of feature output with t-SNE on RML2022 dataset (left: -20dB, center: -6dB, right: 8dB).

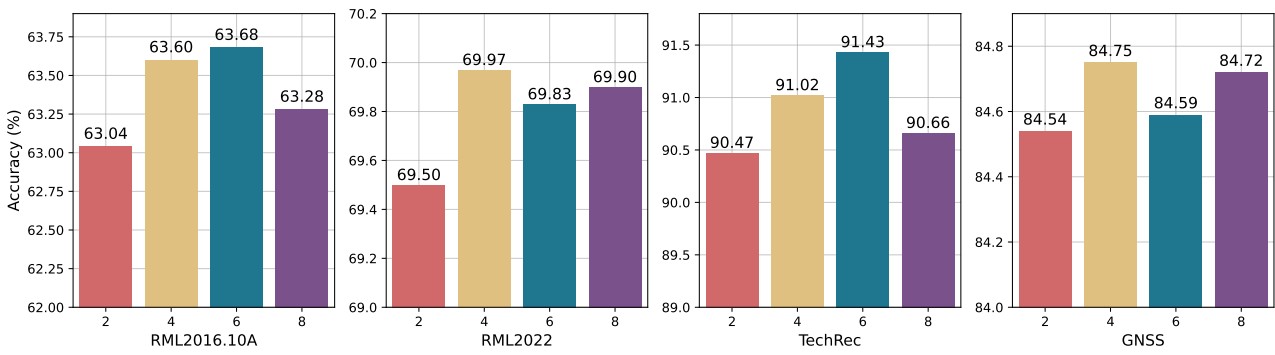

*Figure 10.* The accuracy of different number of layers ranging from 2 to 8 (The dataset from left to right is RML2016.10A, RML2022, TechRec, and GNSS).

*Table 8.* The ablation results on channel parameter learning (percentage), where we present the average and the best accuracy among different SNRs on four datasets.

| Model | RML2016.10A | | RML2022 | | TechRec | | GNSS |
|---|---|---|---|---|---|---|---|
| | Average | Best | Average | Best | Average | Best | Average |
| PWC-Diff (w/o $\phi$) | 61.32 | 91.27 | 68.75 | 97.27 | 89.15 | 99.09 | 84.32 |
| PWC-Diff | **63.60** | **93.91** | **69.97** | **98.64** | **91.02** | **99.23** | **84.75** |

### E.3. Ablation on Channel Parameter Learning

In this section, we present experiments to justify that the channel model $H_\phi(y_0)$ in Equation 6 should be learned from data rather than random. As shown in Table 8, PWC-Diff (w/o $\phi$) denotes a variant where channel parameters in the candidate models are randomly sampled to construct $H$. The results show degraded performance under random parameterization. We believe this is because the random sampling space is too large, and a sufficient number of samples are needed to cover the original channel scenario. This is difficult to achieve within a limited number of epochs, while NNs can learn channel parameters related to $y_0$, thus improving learning effectiveness.

### E.4. The Effect of the Model Size

In this section, we provide experiments with different numbers of encoder layers in FusedFormer. The results are shown in Figure 10. As the number of layers increases, the performance of RML2016.10A, TechRec also improves. For the RML2022 and GNSS dataset, the performance fluctuates with the number of layers. However, when the number of layers

reaches 8, the performance drops. We believe this is because these datasets are relatively small, and the model does not learn thoroughly enough, leading to the performance degradation.

