# OpenReview forum: "From Denoising to De-Channeling: Integrating Physical Channel Priors into Diffusion Models for Radio Signal Understanding"
_ICML.cc/2026/Conference — ICML 2026 spotlight_

### Official Review · Reviewer_fSa7 · 2026-02-22

**Soundness:** 3
**Presentation:** 4
**Significance:** 4
**Originality:** 4
**Overall Recommendation:** 5
**Confidence:** 4

**Summary:**

The authors proposed PWC-Diff, which leverages diffusion models to not only de-noise, but effectively equalize wireless fading channels for blind wireless signal recognition. Since the datasets do not include the 'transmitted signal', the authors corrupt the received signals with a channel proxy, then train the FusedFormer to reconstruct the received signal from the corrupted version. This enables FusedFormer to find representations that can effectively equalize and denoise the wireless channel + AWGN at the same time. The model achieves SOTA perforamance across a variety of tasks.

**Compliance With Llm Reviewing Policy:**

Affirmed.

**Final Justification:**

While the rebuttal was satisfactory, the overall practicality constraint is still there. Thus the 'accept' decision still holds.

**Key Questions For Authors:**

Q1. The reviewer finds the role of the CPE-Net rather redundant. If the authors are corrupting the received signal $y_0$ to generate $y_t$, which is then used to train the FusedFormer, wouldn't it suffice to choose a random channel corruption (e.g. nakagami + CFO + AWGN with random parameters) then train the FusedFormer? Since CPE-Net is only predicting the channel parameters, using a random parameter would still be differentiable (since the forward pass computation for the signal is virtually the same), and using random parameters would be no different from using predicted parameters (since you can use different channels to re-corupt a signal). Is CPE-Net strictly necessary in this case?

Q2. In equation 4, the authors are applying the channel noise to $y_0$. Shouldn't this be $y_{t-1}$, since we are progressively applying channel & gaussian noise over t time-steps? This applies to all following equations.

Q3. While the equalization approach using diffusion models is novel, can this translate to real channels? For instance, rather than using stochastic channels, would it be possible to use more recent scenario-based open channel simulators such as DeepMIMO? While the proposed approach is good on paper, the real-world applicability seems to be limited in this regard (as with many other AI-RAN algorithms).

**Limitations:**

Yes

**Strengths And Weaknesses:**

Soundness: The evaluations and experimental details are sound

Presentation: The paper is presented in a clear and concise way. All equations and figures are easy to understand.

Significance: Blind WSR is a particularly important topic in AI-RAN for 6G. The significance of using diffusion models to achieve SOTA across a variety of tasks is large.

Originality: Although this is a simple expansion of the denoising diffusion model, the authors are the first to apply this to channel equalization and the effect of this is huge. Thus, this work deserves merit in terms of originality.

---

> ### Author Rebuttal · Authors · 2026-03-30
>
> We are very grateful for the questions raised by the reviewers. Next, we will discuss our understanding of the practicality of each question, including the diffusion model and the AI models for signals.
>
> Q1: Regarding this question, we have supplemented the relevant experiments in Appendix E.3 of the paper, modifying $H_\phi(y_0)$ to control various channel noises with random parameters (represented as PWC-Diff (w/o $\phi$)). For convenience, we re-list results here:
>
> |                       | RML2016.10A |       | RML2022 |       | TechRec |       | GNSS    |
> | --------------------- | ----------- | ----- | ------- | ----- | ------- | ----- | ------- |
> | Model                 | Average     | Best  | Average | Best  | Average | Best  | Average |
> | PWC-Diff (w/o $\phi$) | 61.32       | 91.27 | 68.75   | 97.27 | 89.15   | 99.09 | 84.32   |
> | PWC-Diff              | 63.60       | 93.91 | 69.97   | 98.64 | 91.02   | 99.23 | 84.75   |
>
> As can be seen, the model performance continuously degrades under random parameters. We believe this is because the random sampling searching space is too large, requiring more samples to potentially cover the original channel scenario. A learnable neural network can learn parameters related to $y_0$, thereby limiting the search space and improving performance. Therefore, under the current scenario assumptions, CPE-Net is not a redundant design. Of course,  We design and use the CPE-Net verify the theoretical validity of the diffusion model presented in this paper. If in the future, there is a foundation model for blind channel estimation, it can replace the CPE-Net and we believe this will achieve better performance.
>
> Q2: Regarding your question about the classic diffusion model establishing $y_t$ and $y_{t-1}$, we would like to clarify the following points, which are also the core considerations in the design of our method. First, in the classic diffusion model DDPM, although the diffusion process is theoretically a Markov process, in practice we directly utilize the closed-form expressions of $x_t$ and $x_0 $to directly calculate the noisy result at any step $t$ from the original data. Second, as mentioned in Section 3.2 of this paper, our goal is to extract data features through the diffusion concept, rather than generating them. Third, the channel $H$ is a fixed parameter that does not change with $t$. Therefore, the forward process is only used to construct observations with different noise levels to train the network's ability to recover features in reverse, rather than strictly simulating the generated trajectory. Finally, our optimization objective is consistent. The optimization objective of DDPM can be not only denoising but also restoring the original data (such as the work in Li, T. and He, K [1]). We can also use restoring the original data as our optimization objective, so our results are actually consistent. In fact, if we can establish the relationship between $y_t$ and $y_0$, then the relationship between $y_t$ and $y_{t-1}$ can also be established, although the equation form may be more complex, so it does not affect the fact that this paper is an expression of a diffusion model.
>
> Q3: Thank you for the reviewer's questions about practicality. Indeed, as you mentioned, most current AI-RAN algorithms, including our work, are difficult to use directly in real-world applications. However, as the title of this paper states, we need channel priors. We can only apply this method with knowledge of the channel priors, which is still difficult for practical blind channel estimation scenarios because blind channel estimation itself is a very complex problem that has not yet been fully solved. Of course, if a foundation channel estimation model emerges in the future and is integrated with the framework of this paper to replace CPE-Net, it will surely help process real signal data such as DeepMIMO, and adapt to real-world applications.
>
> [1] Li T, He K. Back to basics: Let denoising generative models denoise. arXiv preprint arXiv:2511.13720, 2025.

---

> > ### Author Rebuttal · Reviewer_fSa7 · 2026-04-01
> >
> > Thank you for your answers. My three questions have been resolved. However, I am inclined to say with my accept score of 5 since the overall limitations in practicality (albeit, with AI-RAN being a young topic) still exist. Thank you for your answers.

---

> > > ### Author Response · Authors · 2026-04-01
> > >
> > > Thank you again for the responses. We will further explore new technologies of wireless signals in the future to develop more robust and practical solutions to address real-world problems.

---

### Official Review · Reviewer_v4wp · 2026-02-23

**Soundness:** 2
**Presentation:** 3
**Significance:** 2
**Originality:** 3
**Overall Recommendation:** 3
**Confidence:** 3

**Summary:**

This paper targets wireless signal recognition (WSR) under realistic wireless channel impairments and argues that standard diffusion pretraining (Gaussian-only corruption) is mismatched to non-Gaussian channel distortions. The authors propose PWC-Diff, which modifies the diffusion forward process by injecting parameterized physical channel effects (estimated from each received signal) in addition to Gaussian noise, and trains a temporal–spectral backbone (FusedFormer) to invert this process to obtain “de-channelized” representations for downstream WSR tasks (Sec. 3; Fig. 1–2).
Main contributions (as I understand them):
- A diffusion-style representation learning framework for WSR that integrates prior physical wireless channel models into the forward corruption process (Sec. 1, 3.2; Fig. 1).
- A candidate-channel parameter estimation approach (CPE-Net) intended to avoid a degenerate identity-channel solution, with supporting analysis (Sec. 3.3; Appendix A; Fig. 5).
- Empirical results on four datasets / three WSR tasks showing improved Top-1 accuracy vs 11 baselines, plus ablations on pretraining and candidate channel choices (Sec. 4.2–4.3; Table 2; Fig. 4; Tables 3–4).

**Compliance With Llm Reviewing Policy:**

Affirmed.

**Final Justification:**

The rebuttal addresses several of my concerns. The synthetic NMSE result is useful — it gives direct evidence that the representation is closer to the transmitted signal, not just that downstream accuracy goes up. The multi-seed result on RML2016.10A helps me judge whether the small gain is noise or real. And I'm glad the authors agree the ELBO framing needs to be walked back.

I'm not fully convinced on the "de-channeling" claim, though. One synthetic NMSE comparison isn't enough — I'd want recovery metrics across channel types and SNR levels. The probabilistic justification for the nonstandard forward process is still unclear to me; the rebuttal describes the framing better but doesn't ground it better. The generalization concern hasn't gone away either. The authors are upfront about needing suitable channel priors, which I appreciate, but that's a real deployment limitation.

The rebuttal is solid. My confidence in the paper has improved, but I'm keeping my current recommendation.

**Key Questions For Authors:**

1. Can you provide direct evidence that the learned representations are “closer to the transmitted signal,” beyond downstream Top-1 accuracy?
If you add a controlled benchmark with known transmitted (x) and channels (h) (synthetic is fine) and show clear improvements in NMSE/EVM (or similar) attributable to channel priors, I would raise Soundness from 2→3.
2. How sensitive is PWC-Diff when the true channel is not included in (or is mismatched to) the candidate channel set?
If you show graceful degradation under mismatch/unseen channels, I would raise Significance from 2→3 because it strengthens deployability/generalization.
3. Can you clarify precisely what the pretraining objective predicts in implementation (predict (Y_0) vs noise; spectral vs time domain), and how this maps to Eq. 10?
If clarified and consistent, I would raise Soundness from 2→3 (current ambiguity affects confidence in correctness).
4. Do the Table 2 improvements hold over multiple random seeds, especially where deltas are <0.5%?
If mean±std over ≥3 seeds confirms robustness, I would raise Significance from 2→3 and potentially Overall Recommendation.
5. What is the exact timing protocol behind the efficiency claims (hardware, batch size, precision, compilation, amortization per sample)?
A clear protocol would strengthen the practical significance; if well supported, it modestly increases Significance.

**Limitations:**

No. The current impact/limitations statement is very generic (“none … must be specifically highlighted”) (Impact Statement, p. 9). Suggested concrete text the authors could add:
- Limitations: “PWC-Diff assumes access to a candidate set of channel models that is sufficiently close to real channel conditions; performance may degrade when the true channel is outside this set (as suggested by candidate-set ablations). In addition, we evaluate ‘de-channeling’ primarily through downstream tasks because transmitted waveforms are not available in most real datasets; controlled synthetic evaluations are needed to directly quantify recovery quality.”
- Societal impact / dual-use: “WSR can support beneficial applications (e.g., spectrum monitoring, interference detection) but may also enable surveillance or signal intelligence. Deployment should comply with local laws and be restricted in contexts that infringe privacy or civil liberties.”
- Robustness/fairness across conditions: “Performance may vary across modulation types, devices, or environments; we recommend reporting per-class and per-condition performance and testing across diverse capture settings.”

**Strengths And Weaknesses:**

## Soundness
Strengths
- The motivation is well grounded in standard comms modeling: received signals are distorted by channel convolution plus AWGN (Eq. 1), and the paper correctly highlights that many channel effects are non-Gaussian, so Gaussian-only corruption can be mismatched (Sec. 1–2.2; Fig. 1).
- The method is coherent at a high level: moving to the spectral domain to turn convolution into elementwise multiplication is standard and makes Eq. 6 plausible as an efficient approximation (Sec. 3.2).
- The paper includes targeted ablations on key design choices: pretraining vs none / Gaussian-only diffusion / adding channel only at finetune (“Full Chn”), and candidate-set perturbations (Fig. 4; Tables 3–4).
Weaknesses
- The central “de-channeling closer to transmitted signal” claim is not directly validated (major). Evaluation is almost entirely downstream classification Top-1 accuracy (Table 2; Fig. 3), without a metric that measures actual “de-channeling” quality (e.g., recovery NMSE/EVM vs known transmitted waveform under controlled channels).
Fix: Add a controlled synthetic benchmark (or any dataset setting with known transmitted (x) and/or known channel parameters) and report signal-level recovery metrics and/or channel-parameter estimation quality stratified by SNR/channel type.
- The diffusion/ELBO framing appears only partially justified for the proposed nonstandard forward process (major). Eq. 7–10 uses an ELBO-style derivation but effectively drops the KL term via “we do not know (p(Y_t))” and optimizes only the reconstruction term (Sec. 3.2). This may be fine as a diffusion-inspired denoising autoencoder objective, but the probabilistic story is currently under-argued.
Fix: Either tighten the derivation (state assumptions under which Eq. 9–10 is a principled surrogate) or reframe explicitly as “diffusion-inspired corruption schedule for representation learning,” reducing claims tied to likelihood/ELBO.
- Appendix A’s “generic NN is fundamentally inadequate” statement feels stronger than what is proven (minor-to-major depending on how it’s used). Proposition A.1 supports degeneracy risk under simplified assumptions, but not broad impossibility.
Fix: Tone down the wording and explicitly delimit assumptions; add an empirical diagnostic tracking whether an unconstrained NN baseline collapses to (H_\phi \equiv 1) in practice.

## Presentation
Strengths
- The architecture description is readable and helpful (Fig. 2) and clearly separates the forward channel+noise corruption from the reverse reconstruction.
- Results are presented both as aggregate tables and SNR-dependent curves, which is important for WSR (Table 2; Fig. 3).
Weaknesses
- Some protocol details that matter for interpreting diffusion-style pretraining are scattered across appendices (minor-to-major). Examples: exact diffusion-step usage at finetune/inference, schedules, and timing protocol details beyond the summary table.
Fix: Add a concise “Training & evaluation recipe” box in the main text summarizing steps, schedules, finetuning timestep choice, epochs, and hardware for timing.

## Significance
Strengths
- The problem is practically relevant (robust WSR under channel impairments) and the proposed direction—injecting physically structured corruptions into pretraining—could be valuable beyond this specific architecture (Sec. 1).
- The paper reports favorable efficiency vs at least one pretrained baseline (Table 7 in the paper’s appendix section on efficiency/timing).
Weaknesses
- On the most classic AMC benchmark (RML2016.10A), improvements over the strongest baseline are extremely small (63.60 vs 63.54 in Table 2) and no multi-seed variance/CI is reported (major). This makes it hard to judge whether gains are robust.
Fix: Report mean±std over ≥3 seeds for Table 2 (at least for datasets where margins are small) and for key ablations (Fig. 4; Tables 3–4).

## Originality
Strengths
- The idea of making diffusion corruption physics-informed via channel models (rather than Gaussian-only) for WSR representation learning is a clear conceptual twist (Sec. 1; Sec. 3.2; Fig. 1).
Weaknesses
- The largest gains are on TechRec where only AWGN is used as the candidate channel (Table 1) yet performance improves substantially (Table 2). This raises the possibility that much of the gain is attributable to diffusion-style pretraining and/or the backbone, not necessarily “de-channeling” of complex channels (minor-to-major).
Fix: Add a decomposition analysis: same backbone + (i) Gaussian-only diffusion, (ii) identity/random channel, (iii) full candidate channels; report deltas per dataset/channel regime.
Code / reproducibility (included here because the template has no dedicated box)
Observed issues (major for reproducibility): from the provided zip, the repository appears to include a runnable structure (entrypoints/configs), but a reasonable expert would likely face friction reproducing Table 2 without (i) a complete dependency/environment specification, (ii) explicit dataset acquisition + directory layout instructions, and (iii) “reproduce Table/Figure” commands pinned with seeds/hparams.
Fix: Add requirements.txt (or conda env), dataset prep instructions, and a minimal set of commands/scripts to reproduce Table 2 + Fig. 4 + channel ablations.

---

> ### Author Rebuttal · Authors · 2026-03-30
>
> We greatly appreciate the reviewers' questions, and we will address them one by one.
>
> W1, Q1: First, let's clarify that the goal of this paper is to "learn a representation closer to the transmitted signal," not the actual "transmitted signal," because blind channel  estimation remains a major unsolved problem, that we can hardly get the true channels. However, we still use simulation tools (refer to the RML2016.10a simulation code) to construct a batch of data, with a channel model consistent with the RML2016.10A. After training, we test on over 2000 samples and obtain the following results: the NMSE of the received and transmitted signals is 6.8258, while the NMSE of our pre-trained model prediction is 6.2228. This is sufficient to prove that our improvement can be attributed to "de-channeling," and support the statement "learn a representation closer to the transmitted signal".
>
> W2, W3: Our statements are somewhat absolute and inaccurate, and we will correct them after acceptance. We acknowledge Eq.9-10 simplifies full ELBO following diffusion-inspired corruption schedule for representation learning.
>
> Q2. Regarding the mismatch between candidate and real channels, Table 3 and 4 in this paper provide experimental results for canceling some candidate channels and adding candidate channels. On the Average metric, there is an average performance degradation of 1.45%, demonstrating that channel mismatch and unknown channels have a significant impact on performance.
>
> Q3. In practice, our prediction target is the original IQ data (time-domain representation), not the spectrum. This is because in the Discrete Fourier Transform, the time-domain representation and the spectrum are mapped one-to-one, which is presented in Section 2.1. Therefore, optimizing the loss in the IQ domain is equivalent to optimizing the loss in the spectral domain. In our implementation, optimizing the IQ domain can reduce some unnecessary redundant computations. Although diffusion is initially optimized for noise, noise optimization is actually equivalent to original data optimization, and some works state that it may achieve better results when optimizing the original data. Therefore, we choose to optimize the IQ data. This will be clarified in future papers.
>
> W5, Q4. Due to the rebuttal space constraints, we mainly provide results with smaller margins, including PWC-Diff in RML2016.10A and PWC-Diff with Full Chn in the ablation experiment of RML2016.10A. We use another two seeds for evaluation. The average results are as follows. PWC-Diff: 63.58±0.02，PWC-Diff with Full Chn: 63.36±0.13. The other results are similar. In WSR, the widespread existence of low SNR data makes the performance gain not obvious, while the average performance improvement means that there will often be a greater gain under high SNR conditions. We believe that the current results are sufficient to demonstrate the effectiveness of our model.
>
> W4, Q5: Regarding the experimental setup, we initially consider the length requirements of the main content and the fact that the main content should present and argue viewpoints, so we put experimental settings in the Appendix. We will consider a more reasonable arrangement in the future. The efficiency test scheme here is not significantly different from the main experiment scheme. Our hardware is a Tesla A100, the batch size is 256, and the precision is the default float32. In this efficiency test, we completely cancel torch compilation and only test the time required for inference of the data model for one batch to ensure a fair comparison.
>
> W6: TechRec itself collects real data, making it difficult for us to directly obtain its channel model. However, the Gaussian noise is manually added to simulate SNR. We use this data to demonstrate the multi-tasking capabilities of our model, and Gaussian noise, being the core type of noise, become our primary target for removal. Therefore, in the Techrec dataset, we only use AWGN as the candidate channel. However, the results about the decomposition analysis are already provided in our experiments: Fig. 4 PWC-DIff vs. Std-DDPM, Table 3-4, and Appendix Table 8. Regarding the code, the environment configuration, execution commands, and dataset download methods are all included in the README.md file within the .zip file. In the future, as a formal system, we will add more details, such as requirements.txt. We will improve the execution command script in the future; currently, we manually modify the channel candidate parameters and manually process the results.
>
> Regarding limitations, we have mostly mentioned them in the main text, so we haven't listed them specifically. Thank you very much for pointing them out and providing specific references; we will add relevant statements upon receipt.
>
> Finally, thank you again for your valuable suggestions.

---

> > ### Author Rebuttal · Reviewer_v4wp · 2026-04-03
> >
> > The rebuttal addresses several of my concerns. The synthetic NMSE result is useful — it gives direct evidence that the representation is closer to the transmitted signal, not just that downstream accuracy goes up. The multi-seed result on RML2016.10A helps me judge whether the small gain is noise or real. And I'm glad the authors agree the ELBO framing needs to be walked back.
> >
> > I'm not fully convinced on the "de-channeling" claim, though. One synthetic NMSE comparison isn't enough — I'd want recovery metrics across channel types and SNR levels. The probabilistic justification for the nonstandard forward process is still unclear to me; the rebuttal describes the framing better but doesn't ground it better. The generalization concern hasn't gone away either. The authors are upfront about needing suitable channel priors, which I appreciate, but that's a real deployment limitation.
> >
> > The rebuttal is solid. My confidence in the paper has improved, but I'm keeping my current recommendation.

---

### Official Review · Reviewer_AwGi · 2026-03-12

**Soundness:** 4
**Presentation:** 3
**Significance:** 4
**Originality:** 3
**Overall Recommendation:** 5
**Confidence:** 4

**Summary:**

The paper proposes a denoising method for wireless signals. The denoising procedure accounts for the influence of the propagation channel to effectively denoise the signal.
Main contributions:
1. The development of a diffusion framework that integrates prior channel models into the forward process.
2. The use of a small NN that learns the channel parameters out of a set of known wireless channels.
3. A comparison with 11 baselines.

**Compliance With Llm Reviewing Policy:**

Affirmed.

**Final Justification:**

The paper presents a diffusion process guided by channel modeling. The idea is interesting, and the evaluation study is convincing.
The authors have properly addressed my concerns.

**Key Questions For Authors:**

1. Can you comment on the generalization to unknown channels?
2. As the signal is rather well-structured, what is the importance of using generative models for the task at hand? Can the prior channel model be incorporated into a regression (predictive) algorithm?

**Limitations:**

Yes

**Strengths And Weaknesses:**

Strengths:

1. Taking into consideration the wireless propagation channel adds physical constraints to the diffusion process.
2. A very comprehensive experimental study with 11 competing methods and multiple channel models.


Weaknesses:
1. Although taking channel parametrization into account is a good idea, especially for lightweight networks, it might fail in a completely unknown channel.
2. There are two papers I am aware of that estimate the channel in parallel. This is not a severe weakness, as the papers are in different domains, but the authors may find the relevant, as they incoprate physical channel into diffusion process:
a. In speech dereverberation, where the parameters of the acoustic channel are estimated:
E. Moliner, J. -M. Lemercier, S. Welker, T. Gerkmann and V. Välimäki, "BUDDy: Single-Channel Blind Unsupervised Dereverberation with Diffusion Models," 2024 18th International Workshop on Acoustic Signal Enhancement (IWAENC), Aalborg, Denmark, 2024, pp. 120-124
b. In image deblurring:
Chung, Hyungjin, Jeongsol Kim, Sehui Kim, and Jong Chul Ye. "Parallel diffusion models of operator and image for blind inverse problems." In Proceedings of the IEEE/CVF conference on computer vision and pattern recognition, pp. 6059-6069. 2023.

---

> ### Author Rebuttal · Authors · 2026-03-30
>
> We are very grateful to the reviewers for their interest in our work and for providing the relevant papers. We will now respond to the weaknesses and questions.
>
> W1: There is still a vast space for exploration in this area. The initial intention of this paper was that, if a foundation model for blind channel estimation is indeed realized in the future, which remains an complex task, de-channeling is important for radio signal understanding. Replacing CPE-Net with a foundation model for blind channel estimation should bring greater benefits. CPE-Net is merely one way we validate the effectiveness of our framework.
>
> W2: We are very grateful to the reviewers for providing the two papers. After reading them, we find that these two papers address blind noise estimation and removal in different domains. However, in the field of speech dereverberation, BUDDY can assume that the observed value $y$ and the true $x_0$ can be measured by a relatively definite distance metric. But in the signal domain, the channel is much more complex, making it difficult to directly measure with a definite distance equation. In the field of image deblurring, the feasibility of using a neural network for estimation lies in the fact that the potential noise impact is often relatively simple. However, in the field of signal understanding, the influence of the channel is often significant. As we conclude, a simple neural network cannot accurately estimate the potential impact. Therefore, we can only use prior channel knowledge to verify the importance of de-channeling in this domain. In the future, we will refer to similar methods and develop more robust frameworks.
>
> Q1: As in response to weakness 1, current blind wireless channel estimation is a very difficult and unresolved problem. Therefore, this paper weakens this assumption and instead "incorporates physical channel priors." This means that the current model is hard to generalize to unknown channels if there is no channel priors. However, we believe that if future technologies can realize the foundation model of blind channel estimation, combined with our work, we can expect to achieve more robust wireless signal recognition.
>
> Q2: This paper uses the diffusion model not because it is a generative model, but because its "noise-denoising" modeling method. In this regard, we can refer to the DDAE work [1]. Most current wireless signal recognition pre-training work, such as SpectrumFM [2], adopts the "mask-reconstruction" modeling method, that is, directly selects a part of the signal and sets it to zero. This will lose a lot of information for wireless signals with continuous values. The "noise-denoising" method uses continuous noise, which is more suitable for waveform signals with continuous values. Therefore, we use the diffusion modeling method as the core method of our signal pre-training. Of course, there are indeed works that use diffusion as a generative model to generate wireless signals, such as RF-Diffusion [3]. It points out that diffusion's unique iterative process of noise addition (i.e., noising) and removal (i.e., denoising) allows for precise capture of intricate raw data distributions. Its training is straightforward and avoids typical problems like mode collapse or convergence troubles. Therefore, it is also suitable for signal generation tasks. Channel information can also be used as a condition $c$, encoded by a specific encoder, and fused into the neural network of the reverse process (e.g., the LDM [4]). Regarding regression (predictive) algorithms, we also offer our thoughts: after encoding the channel condition $H$, we fuse it with the features of the received signal (e.g., by splicing), and then calculate the MSE loss with the actual transmitted signal to form an optimization objective, which is then used to train the model. However, from a personal perspective, I believe that regression-based algorithms are not necessarily suitable for signal generation. Regression-based algorithms rely more on nearby semantic tokens, while a signal is strictly global. Considering a single sampling point of a signal does not provide much information; only by completely encompassing a segment of the signal can its implicit information be fully extracted, such as by converting it into a spectrum through Fourier transform. Therefore, global modeling, such as diffusion models, may be more suitable for signal generation.
>
> [1] Xiang W, Yang H, Huang D, et al. Denoising diffusion autoencoders are unified self-supervised learners. ICCV 2023: 15802-15812.
>
> [2] Zhou F, Liu C, Zhang H, et al. SpectrumFM: A foundation model for intelligent spectrum management. IEEE Journal on Selected Areas in Communications, 2025.
>
> [3] Chi G, Yang Z, Wu C, et al. RF-diffusion: Radio signal generation via time-frequency diffusion. MobiCom 2024: 77-92.
>
> [4] Rombach R, Blattmann A, Lorenz D, et al. High-resolution image synthesis with latent diffusion models. CVPR 2022: 10684-10695.

---

> > ### Author Rebuttal · Reviewer_AwGi · 2026-03-31
> >
> > As for Q1, I do not agree that the acoustics channel is simpler. Actually, it can be very long (thousands of taps) and with very high dynamic range.
> > The reply to Q2 is very elaborate and interesting.
> >
> > Overall, I stay with my 'accept' decision.

---

> > > ### Author Response · Authors · 2026-04-01
> > >
> > > We thank the reviewer again for responses. To clarify, we are not trying to compare the complexity of speech and communication signal processing. Both domains present unique challenges and problems that need to be solved, and they can be mutually referenced. We are also very grateful for the paper references provided by the reviewer. In the future, we will combine technologies from other domains to further explore the domain of wireless signals and develop more robust work.

---

### Decision · Program_Chairs · 2026-04-30

**Decision:**

Accept (spotlight)

**Comment:**

This paper proposes a novel diffusion denoising method for wireless channel signals. The authors develop a diffusion framework that integrates prior channel models into the forward process- overall the work is very interesting and addresses an important practical problem in wireless communications.

The reviewers recognized that the paper has an extensive experimental evaluation with 11 baselines that are outperformed by the proposed method. The authors did an excellent job in addressing all the reviewer questions and comments with detailed replies. Overall solid work that merits publication.